# SpinVLA: A Spectral-Invariant Vision-Language-Action Model for Robotic Manipulation

## Abstract

Vision-language-action (VLA) models trained on large-scale robot demonstration datasets have achieved impressive in-distribution performance, yet they can fail catastrophically under minor domain shifts. For instance, a VLA-trained robot tasked to "pick the red block" may flounder due to various environmental disturbances such as lighting changes or scene clutter. To address this limitation, we propose SpinVLA, a novel end-to-end VLA architecture that leverages the mathematical equivalence between spectral decomposition and contrastive learning to improve robustness. Drawing inspiration from causal inference principles, which suggest that stable features persist across environments, we hypothesize that consistent patterns in successful demonstrations represent task-relevant information rather than spurious correlations, i.e., statistical associations unrelated to the true causal factors of task performance. Our approach integrates spectral decomposition to identify demonstration-consistent features, contrastive learning to enforce representational stability, and efficient low-rank adaptation modules for environment-specific tuning. Extensive experiments using the open-source LIBERO datasets show that SpinVLA significantly improves robotic manipulation task success rates compared to baseline VLAs under visual perturbations and the presence of out-of-distribution objects, while maintaining comparable in-distribution performance.

## 1 Introduction

The deployment of robots in unstructured real-world environments demands systems that generalize beyond their training distributions. Vision-language-action (VLA) models have emerged as a promising paradigm, leveraging large-scale pretraining to enable robots to follow natural language instructions across diverse scenarios. Models such as RT-2 (Zitkovich et al., 2023) and OpenVLA (Kim et al., 2025), and recent variants like TinyVLA (Wen et al., 2025) and EfficientVLA (Yang et al., 2025), demonstrate impressive capabilities in controlled settings. Nevertheless, their performance drastically degrades when confronted with distribution shift, spurious visual correlations, or ambiguous instructions. This is not merely a technical limitation, it is a fundamental barrier to deploying autonomous robots in human environments where safety, reliability, and interpretability are paramount (Diehl & Ramirez-Amaro, 2022; Liu et al., 2023b).

Consider a VLA-equipped robot trained on manipulation tasks where thousands of teleoperated trajectories demonstrate basic pick-and-place operations. These trajectories can exhibit systematic biases from the collection protocols. For example, objects picked from the left side of the workspace will consistently use slower, arc-like motions due to the camera placement and operator viewing angles, while objects from the center will follow more direct paths. The robot learns to associate spatial positions with trajectory shapes rather than understanding that the actual determinants are obstacle avoidance, gripper approach constraints, or object geometry. When deployed with a different camera configuration or workspace layout where these spatial correlations no longer hold, the robot will inevitably attempt inappropriate trajectories (e.g., the model executes wide arcs when direct paths would suffice or vice versa). This is due to the model capturing spurious correlations between workspace positions and motion patterns that emerged from consistent data collection setups rather than learning the underlying geometric and physical constraints that govern the trajectory planning.

**Inputs**  **SpinVLA**  **Output**

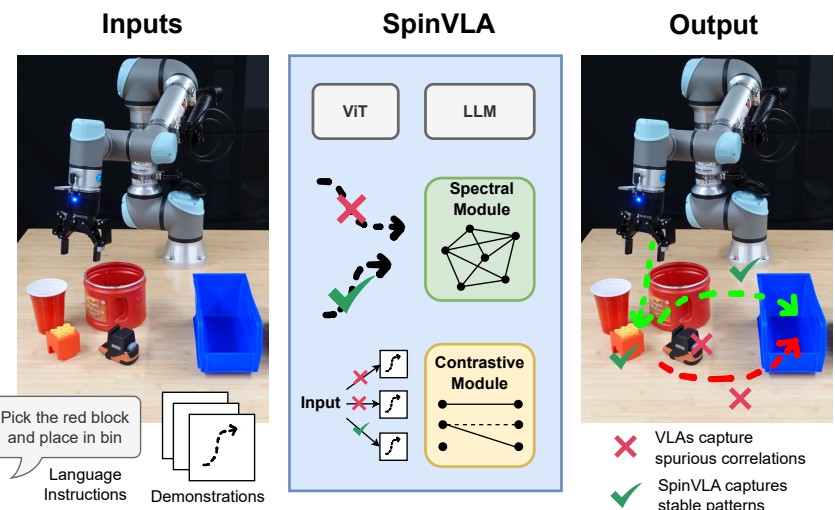

Figure 1: An overview of SpinVLA. Given visual inputs, language instructions, and training demonstrations, baseline VLAs capture spurious correlations and fail under distribution shifts. Conversely, SpinVLA integrates spectral and contrastive modules to capture stable patterns, enabling robust robotic manipulation.

This brittleness stems from a fundamental limitation in how VLAs process the demonstration data (Florence et al., 2022; Shafiullah et al., 2022). These models excel at capturing statistical associations present in the training distributions, but they struggle to distinguish stable task-relevant patterns from transient correlations. During training on large-scale datasets, VLAs capture spurious correlations arising from data collection biases. These correlations associate specific environmental conditions with successful grasps, workspace positions with trajectory shapes, or verbal phrasings with action sequences. The multimodal nature of VLA inputs compounds this problem. Visual observations mix task-relevant information such as the object's geometry and material properties with irrelevant confounders like the background and camera angle, while language instructions blend essential commands with stylistic variations and ambiguous references.

The challenge of separating relevant from irrelevant features in demonstrations is particularly difficult for robotic manipulation tasks. Unlike robot navigation with discrete, reversible decisions, manipulation requires continuous, precise control where errors compound irreversibly since dropped objects shatter and slipped grasps cannot be retroactively corrected. This asymmetry in error recovery, combined with the need to disentangle relevant factors across visual, linguistic, and action modalities, makes robust pattern understanding crucial for safe manipulation. While full causal discovery may be intractable in high-dimensional VLA settings (Pearl, 2009; Peters et al., 2017), recent advances reveal that spectral methods naturally discover stable representations through implicit decomposition of the data manifold (HaoChen et al., 2021; Tan et al., 2024; Alshammari et al., 2025). These mathematical equivalences suggest that we can bias models toward stable patterns via architectural design rather than requiring explicit causal modeling, thus motivating our use of spectral regularization to improve VLA stability.

In summary, we introduce SpinVLA (SPectral-INvariant VLA), a framework inspired by causal principles but implemented through efficient regularization methods (Figure 1). Our approach builds on the hypothesis that patterns consistent across successful demonstrations are more likely to represent task-relevant information instead of data collection artifacts. We make the following key contributions.

- **Spectral-contrastive decomposition**: We leverage the mathematical equivalence between spectral clustering and contrastive learning to bias learning toward demonstration-consistent patterns via efficient computation.

- **Multi-scale regularization**: We apply regularization at the action and sequence levels to discourage spurious correlations while preserving demonstrated behaviors.

- **Efficient adaptation**: We extend low-rank adaptation with spectral regularization, enabling rapid environment-specific tuning while maintaining learned demonstration patterns.

Our model and source code are publicly available at SpinVLA.

## 2 RELATED WORK

### 2.1 VISION-LANGUAGE-ACTION MODELS

VLAs have emerged as a promising paradigm for robotic control by unifying perception, language understanding, and action generation. RT-2 (Zitkovich et al., 2023) pioneered web-scale pretraining to support manipulation tasks, although its 55B parameters restrict real-time deployment. Subsequent work has focused on efficiency with OpenVLA (Kim et al., 2025) achieving comparable performance using 7B parameters through pretrained encoders. TinyVLA (Wen et al., 2025) and Fast (Pertsch et al., 2025) further improved inference speed via knowledge distillation and action tokenization, respectively. These improvements have enabled a broader deployment of VLAs on resource-constrained robotic systems.

Beyond computational efficiency, related work has explored architectural innovations to improve robustness and expressiveness. Action chunking (Zhao et al., 2023) and diffusion-based policies (Chi et al., 2024; Black et al., 2024) have shown remarkable success in dexterous manipulation tasks by modeling richer action distributions. Helix (Figure AI Team, 2025) adopted a dual-system approach that separates slow semantic reasoning from fast reactive control. Foundation models like RoboCat (Bousmalis et al., 2024) have demonstrated self-improvement capabilities through continual learning. However, existing VLAs learn statistical associations from observational data and conflate causal relationships with spurious correlations (Ma et al., 2024).

### 2.2 CAUSAL INFERENCE FOR ROBOTICS

Pearl's causal hierarchy provides a framework for understanding different levels of reasoning, from associational to interventional and counterfactual. While deep learning typically operates at the associational level, other work has attempted to incorporate causal principles. For instance, IRM (Arjovsky et al., 2019) learns predictors invariant across environments and causal representation learning (Schölkopf et al., 2021) discovers latent causal variables. In robotics, Diehl & Ramirez-Amaro (2023) predicted failures through causal analysis, Lee et al. (2023) applied causal reasoning to enable skill transfer in simulation, and Cannizzaro et al. (2024) used probabilistic programming to enable interventional inference. Nonetheless, these approaches require either explicit causal graphs or environment labels that are unavailable in robotic manipulation settings where physical interactions create complex, partially-observable causal structures.

### 2.3 SPECTRAL CLUSTERING AND CONTRASTIVE LEARNING

The mathematical connection between spectral clustering and contrastive learning gives powerful tools for discovering smooth representations on data manifolds. The foundational work by HaoChen et al. (2021) proved that minimizing the InfoNCE loss implicitly results in spectral decomposition on similarity graphs. This equivalence was extended by Tan et al. (2024), who showed that contrastive objectives correspond to Laplacian eigenmaps. Li et al. (2021) introduced contrastive clustering methods, while Williams & Robles-Kelly (2025) combined online spectral clustering with contrastive learning for unsupervised classification.

The I-Con framework (Alshammari et al., 2025) unified these insights by showing different contrastive losses are special cases of a general information-theoretic bound, while recent work by Wang et al. (2024) has explored semantic spectral clustering with contrastive learning and neighbor mining techniques. Although prior work uses spectral properties as training objectives, we employ spectral decompositions computed from demonstration trajectories to regularize action predictions, biasing the model toward patterns that appear consistently in successful demonstrations. This distinction is crucial for robotics, where adaptation must occur efficiently as robots encounter novel environments.

## 2.4 Distribution Shift in Robotic Manipulation

Distribution shift poses fundamental challenges for robotic manipulation. Past approaches rely on domain randomization (Tobin et al., 2017) to vary visual parameters during training or meta-learning (Finn et al., 2017) to enable rapid adaptation. However, extensive randomization can degrade in-distribution performance, while meta-learning requires diverse training environments that may be unavailable. The CausalWorld benchmark (Ahmed et al., 2021) demonstrated that current methods fail under causal perturbations that alter data-generating processes rather than surface statistics.

Our methodology fundamentally differs from prior work that either assumes access to ground-truth causal graphs or performs expensive causal discovery. We provide an orthogonal solution by regularizing the model toward patterns consistent across demonstrations, achieving robustness through architectural bias rather than requiring diverse training data or explicit adaptation mechanisms. We adopt a pragmatic stance, acknowledging that perfect causal identification may be impossible in complex robotic settings. Instead, we focus on architectural innovations that are inspired by causal principles, yet carried out through proven spectral-contrastive equivalences, to realize stable patterns from demonstrations.

## 3 Methodology

We present SpinVLA, an architecture that combines spectral decomposition with contrastive learning to improve VLA robustness against distribution shifts in robotic manipulation scenarios. The problem is first formulated in Section 3.1. Section 3.2 provides an overview of the architectural components. The technical details of the model are described in Sections 3.3- 3.6 Finally, in Section 3.7, we present SpinVLA's training objective function.

### 3.1 Problem Formulation

Consider a VLA, $f_\theta : \mathcal{X} \to \mathcal{A}$, that maps multimodal inputs $\mathcal{X} = \mathcal{V} \times \mathcal{L} \times \mathcal{H}$ consisting of visual observations $\mathcal{V}$, natural language instructions $\mathcal{L}$, and action histories $\mathcal{H}$, to actions $\mathcal{A} \subset \mathbb{R}^d$. Given a demonstration dataset of state-action pairs $\mathcal{D} = \{(x, a)\}$, standard behavioral cloning (BC) minimizes

$$\mathcal{L}_{\text{BC}} = \mathbb{E}_{(x,a)\sim\mathcal{D}}[\|f_\theta(x) - a\|^2]. \tag{1}$$

This objective captures all statistical associations present in $\mathcal{D}$, including spurious correlations introduced by collection biases. From a causal perspective, such spurious correlations arise from confounding factors that create non-causal associations between inputs and actions. Rather than attempting explicit causal discovery, which is often intractable in high-dimensional settings, our goal is to bias learning toward patterns that consistently appear across successful demonstrations via architectural regularization.

### 3.2 Architecture Overview

Building upon the standard VLA framework identified in the problem formulation, our architecture enhances VLAs with three major components that work synergistically to combat spurious correlations. First, spectral projections computed from demonstration trajectories guide action predictions to the manifold of successful behaviors. Second, contrastive heads learn smooth representations via InfoNCE optimization, encouraging consistency across augmented views of the same task. Third, low-rank adaptation (LoRA) modules with spectral regularization enable efficient environment-specific tuning while preserving demonstration-consistent patterns. These components integrate at the action generation and temporal sequencing levels, maintaining the base VLA's computational efficiency. Figure 2 shows an overview of the SpinVLA implementation.

### 3.3 Spectral Decomposition via Contrastive Learning

Having established the architectural components, we now describe how contrastive learning induces spectral decomposition to identify task-relevant patterns. Following the contrastive learning framework of Oord et al. (2018), the InfoNCE objective is widely used to learn representations from positive and negative sample pairs. HaoChen et al. (2021) further demonstrated that minimizing

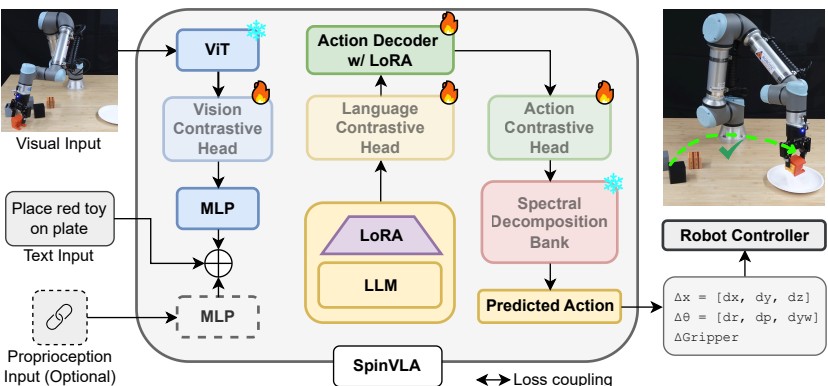

Figure 2: The SpinVLA architecture. Visual inputs first pass through a frozen vision transformer (ViT) encoder and trainable projection layer before joining text tokens in the large language model (LLM) with LoRA adapters. During training, contrastive heads (semi-transparent) compute InfoNCE losses for multimodal alignment. A precomputed spectral decomposition bank constrains actions to demonstration patterns, with both components providing gradients without affecting inference. The action decoder outputs robot manipulator delta commands.

this objective induces embeddings aligned with spectral decompositions on similarity graphs, effectively identifying smooth variations in the data through efficient optimization. This connection is fundamental to our approach, as it provides the theoretical foundation for why the contrastive heads (Figure 2) discover stable patterns across demonstrations. The InfoNCE objective is defined as

$$\mathcal{L}_{\text{InfoNCE}} = -\mathbb{E}_{(x,x^+)} \left[ \log \frac{\exp(\phi(x) \cdot \phi(x^+)/\tau)}{\sum_{x^-} \exp(\phi(x) \cdot \phi(x^-)/\tau)} \right],$$ (2)

where $\phi(x)$ denotes the encoder representation of input $x$ and $\tau$ is a temperature parameter. Prior work has shown that minimizing this loss encourages embeddings that approximate eigenvectors of the similarity graph, revealing principal modes of variation. We hypothesize that these modes correlate with task-relevant patterns when computed from successful demonstrations.

To apply this theoretical insight, we design augmentations that create meaningful positive pairs while preserving semantic task features. Visual augmentations (color jittering, spatial crops) maintain object relationships while varying lighting and viewpoint. Language augmentations use paraphrasing (e.g., "pick up the red block" to "grab the red cube") to express identical goals through different linguistic forms. Action augmentations add Gaussian noise within successful trajectory bounds, similar to diffusion policies (Chi et al., 2024), creating slightly different but equally valid paths. By defining positive pairs as augmented versions of the same demonstration and negative pairs as samples from different demonstrations, the contrastive objective learns to distinguish task-relevant invariances from spurious variations.

### 3.4 ACTION TRAJECTORY ANALYSIS

While Section 3.3 established how contrastive learning uncovers the spectral embeddings of multimodal inputs through the contrastive heads, robustness in robotic manipulation also depends on analyzing the action space directly through the spectral decomposition bank component, as displayed in Figure 2. This complementary analysis operates on a different level than the contrastive heads by examining the geometric structure of demonstrated trajectories themselves. Demonstrations often encode consistent motion patterns that reflect task-relevant constraints such as obstacle avoidance or gripper-approach strategies, however they may also contain collection-specific artifacts. To disentangle and isolate stable task-relevant structures, we analyze the similarity of the demonstrated trajectories via spectral decomposition of their alignment structure. This provides a principled basis for identifying the low-dimensional subspaces of consistent action patterns across successful demonstrations, which is then used by the spectral decomposition bank to constrain the action decoder's outputs.

To construct this spectral representation, we must first measure the similarity between trajectories of potentially different lengths. Concretely, we construct an affinity matrix based on dynamic time warping (DTW), which measures alignment between variable-length trajectories. Since DTW does not satisfy the triangle inequality, the standard Gaussian kernel $\exp(-d^2/2\sigma^2)$ is not guaranteed to produce a positive semi-definite (PSD) matrix. To mitigate this dilemma, we employ a regularized affinity matrix formulation,

$$W_{ij} = s_i \cdot s_j \cdot \exp\left(-\frac{d_{\mathrm{DTW}}(\tau_i, \tau_j)}{2\sigma^2}\right) + \epsilon\delta_{ij}, \tag{3}$$

where $s_i \in \{0, 1\}$ indicates whether trajectory $i$ succeeded (so only pairs of successful trajectories contribute), $\tau_i = (a_1^i, \ldots, a_T^i)$ denotes the action sequence, $d_{\mathrm{DTW}}$ is the DTW distance, $\epsilon > 0$ is a regularization parameter (typically $10^{-6}$), and $\delta_{ij}$ is the Kronecker delta. The regularization term $\epsilon\delta_{ij}$ improves numerical stability and shifts eigenvalues upward, alleviating issues caused by the non-PSD kernel.

In practice, we use FastDTW (Salvador & Chan, 2007) with a Sakoe-Chiba band radius of 10 for computational efficiency, reducing complexity from $O(n^2)$ to $O(nr)$ while maintaining accuracy. The bandwidth parameter $\sigma$ is automatically computed as the median of non-zero pairwise distances among successful trajectories, providing adaptive scaling based on the demonstration data distribution. By eigendecomposition of this regularized affinity matrix, we identify the principal modes of variation in successful demonstrations. We then compute the normalized Laplacian $L = I - D^{-1/2}WD^{-1/2}$, where $D$ is the degree matrix, and extract its eigenvectors. The spectral gap in the eigenvalues reveals a natural separation and we select $k^*$ components before the largest gap, typically capturing 85% of demonstration variance. These eigenvectors define a projection operator $\Pi_{k^*} : \mathbb{R}^{T \times d} \to \mathbb{R}^{k^*}$ that maps trajectory vectors to a lower-dimensional subspace. This projection operator becomes the core component of the spectral decomposition bank, providing a precomputed regularization term that encourages action predictions to remain near demonstration patterns during both training and adaptation.

## 3.5 Multi-Level Regularization

Having established spectral representations through contrastive learning and trajectory analysis, we now add regularization. Spurious correlations can emerge across the model hierarchy, from low-level visual features to high-level action planning. We address this by regularizing action generation via the spectral decomposition bank and temporal sequencing trajectory consistency losses, biasing the model toward patterns consistent with successful demonstrations.

**Action-level regularization.** The spectral decomposition bank constrains predicted actions to the subspace of demonstrated patterns using a consistency loss. For a sequence of predicted actions $\hat{\tau} = (f_\theta(x_1), \ldots, f_\theta(x_T))$ from the action decoder, we compute

$$\mathcal{L}_{\mathrm{consist}} = \mathbb{E}_{\tau \sim \mathcal{D}}\left[\|vec(\hat{\tau}) - \Pi_{k^*}(vec(\hat{\tau}))\|^2\right], \tag{4}$$

where $vec(\cdot)$ concatenates the trajectory into a vector compatible with the projection operator $\Pi_{k^*}$ (Section 3.4). This loss penalizes trajectories that deviate significantly from the subspace of demonstrated patterns. The projection can be precomputed and cached, adding trivial overhead during inference.

**Temporal-level regularization.** Beyond spatial constraints, robot actions must evolve smoothly to reflect the temporal dynamics observed in demonstrations. Without this, VLAs tend to overfit to frame-level correlations and produce incoherent motion. We regularize the temporal derivatives of predicted action sequences using

$$\mathcal{L}_{\mathrm{temporal}} = \sum_{t=1}^{T-1} \|(f_\theta(x_{t+1}) - f_\theta(x_t)) - \bar{v}_t\|^2, \tag{5}$$

where $\bar{v}_t$ represents the mean velocity at timestep $t$ computed from successful demonstrations. This prevents jerky, unstable movements that risk damaging objects or the robot itself. Together, action-level regularization keeps predictions within the manifold of demonstrated trajectories, while temporal regularization enforces a smooth evolution consistent with the robot's dynamics. These complementary components integrate with the contrastive losses and BC objective in the overall training framework.

| Method | LIBERO-Spatial | LIBERO-Object | LIBERO-Goal | LIBERO-Long |
|---|---|---|---|---|
| OpenVLA (published) | 84.7 | 88.4 | 79.2 | 53.7 |
| OpenVLA (our replication) | 78.8 | 80.5 | 70.1 | 48.9 |
| TinyVLA (our implementation) | 64.8 | 55.9 | 61.3 | 27.2 |
| SpinVLA (ours) | 75.2 | 71.6 | 73.4 | 35.1 |
| Improvement vs. our OpenVLA | -3.6 | -8.9 | +3.3 | -13.8 |
| Improvement vs. our TinyVLA | +10.4 | +15.7 | +12.1 | +7.9 |

Table 1: Success rates (%) on the LIBERO Liu et al. (2023a) dataset benchmarks. The published OpenVLA results from the original paper are shown for reference. Our attempts to replicate the official fine-tuning instructions yielded suboptimal performance for both OpenVLA and TinyVLA baselines. All the models in the experiments were fine-tuned with LoRA (rank=32) using identical procedures.

### 3.6 EFFICIENT ADAPTATION VIA SPECTRAL-CONSTRAINED LoRA

The regularization strategies described thus far enhance training robustness, yet practical deployment often requires adapting to new environments with limited data. To address this concern, we extend LoRA with spectral regularization, enabling efficient fine-tuning while preserving the demonstrated patterns learned during pretraining. In standard LoRA (Hu et al., 2022), weight updates decompose to $W = W_0 + BA$, where $W_0$ is the frozen pretrained matrix and $BA$ is a low-rank update. We extend standard LoRA by incorporating the spectral constraints from our trajectory analysis directly into the adaptation process. Specifically, we regularize the gradient updates during LoRA fine-tuning to remain close to the demonstrated subspace identified by the spectral decomposition bank. During adaptation, we project the gradients of the LoRA parameters onto the spectral subspace derived from demonstrations prior to each update. This is implemented by modifying the gradient $\nabla_{BA}\mathcal{L}$ to $\Pi_{k^*}(\nabla_{BA}\mathcal{L})$ before the parameter update, where $\Pi_{k^*}$ is the same projection operator computed in Section 3.4. This projection adds negligible computational cost, requiring only a single matrix multiplication. The spectral constraint ensures that even when adapting to new environments with different visual appearances or object configurations, the model maintains the fundamental motion patterns and task structures learned from the original demonstrations.

### 3.7 TRAINING OBJECTIVE FUNCTION

We now present the training objective function that combines the components of SpinVLA and their associated losses. The complete training objective balances learning from demonstrations with regularization toward consistent patterns,

$$\mathcal{L} = \mathcal{L}_{\text{BC}} + \lambda_1 \mathcal{L}_{\text{InfoNCE}} + \lambda_2 \mathcal{L}_{\text{consist}} + \lambda_3 \mathcal{L}_{\text{temporal}}, \tag{6}$$

where the hyperparameters control the relative importance of each component. The BC loss $\mathcal{L}_{\text{BC}}$ provides the primary learning signal via demonstrations, while $\mathcal{L}_{\text{InfoNCE}}$ from the contrastive heads learns robust multimodal representations. The consistency loss $\mathcal{L}_{\text{consist}}$ enforces spectral regularization through the decomposition bank, and $\mathcal{L}_{\text{temporal}}$ ensures smooth action sequences.

## 4 EXPERIMENTS AND RESULTS

We conducted a comprehensive set of experiments to evaluate SpinVLA's effectiveness in addressing distribution shift and spurious correlations on robotic manipulation tasks. The evaluation spans simulation benchmarks, experiments with a real robot arm, and a detailed ablation study that reveals the critical importance of each architectural component.

### 4.1 ROBOTIC MANIPULATION RESULTS

Table 1 presents success rates across the LIBERO (Liu et al., 2023a) datasets. SpinVLA demonstrates strong performance by achieving competitive results, but using a much smaller language model than OpenVLA, making it a compelling option for resource-constrained deployment scenarios. Despite utilizing a 1.5B parameter LLM similar in scale to TinyVLA, SpinVLA substantially

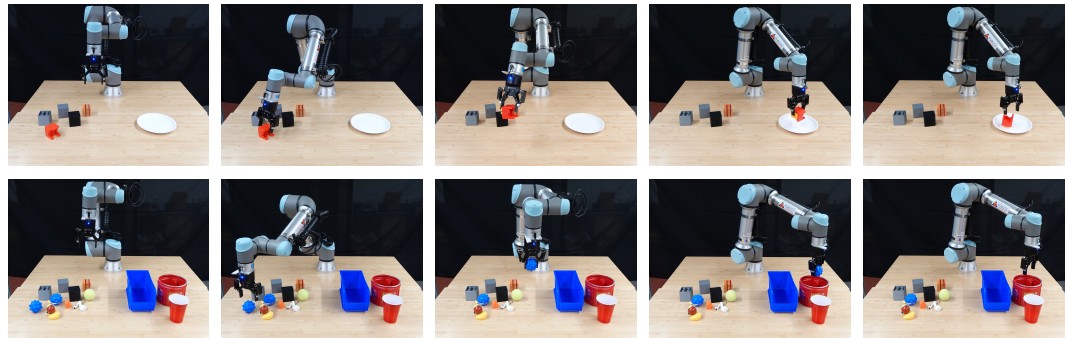

Figure 3: Top row: A baseline demonstration of transferring from the LIBERO simulation environment to real-world task execution. Bottom row: A purposefully cluttered scene to test model robustness against confusing prompts and visual distractions.

| Configuration | LIBERO-Spatial | LIBERO-Object | LIBERO-Goal | LIBERO-Long |
|---|---|---|---|---|
| SpinVLA (Full Model) | **75.2 ± 1.3** | **71.6 ± 0.9** | **73.4 ± 1.1** | **35.1 ± 1.4** |
| w/o InfoNCE ($\lambda_1 = 0$) | 70.5 ± 1.0 | 67.8 ± 1.5 | 69.3 ± 1.2 | 31.5 ± 1.1 |
| w/o Consistency ($\lambda_2 = 0$) | 71.6 ± 1.4 | 68.4 ± 1.0 | 69.9 ± 1.3 | 31.8 ± 0.8 |
| w/o Temporal ($\lambda_3 = 0$) | 74.3 ± 1.1 | 70.2 ± 1.2 | 71.7 ± 0.9 | 34.4 ± 1.3 |
| w/o Any Regularization | 70.1 ± 3.8 | 67.1 ± 2.5 | 68.6 ± 2.3 | 30.8 ± 4.1 |

Table 2: An ablation study showing the performance degradation when individual components are disabled. Removing any component eliminates any gains over the baseline model.

outperforms TinyVLA across all four LIBERO benchmark tasks with impressive gains ranging from 7.9 to 15.7 percentage points, showcasing superior architectural design and training strategies within the same model size class. Most notably, SpinVLA achieves a 3.3 percentage point improvement over our OpenVLA replication on the challenging LIBERO-Goal task, demonstrating that careful model design can partially compensate for reduced model capacity in certain scenarios.

Even though the smaller LLM limits SpinVLA's expressive power compared to OpenVLA's larger backbone resulting in expected performance gaps on some tasks, SpinVLA successfully bridges much of the performance difference between lightweight and full-scale models, achieving 75.2%, 71.6%, 73.4%, and 35.1% success rates across the four benchmarks. These results establish Spin-VLA as an effective middle-ground solution that balances computational efficiency with task performance. This makes it particularly valuable for applications where deployment constraints prohibit the use of larger models, yet substantially better performance than minimal baseline implementations like TinyVLA is required.

## 4.2 EXPERIMENTS WITH A ROBOT MANIPULATOR

We deployed our trained model on a Universal Robots UR5e manipulator for real-world validation. As illustrated in Figure 3 (top row), SpinVLA successfully transfers policies learned within the LIBERO simulation to physical task execution, performing precise pick-and-place operations with various objects including blocks, cups, and containers. The model demonstrates robust performance even in challenging cluttered environments, Figure 3 (bottom row), where it accurately identifies and manipulates target objects despite visual distractors and ambiguous arrangements. Notably, the robot maintains task accuracy when faced with multiple similar-looking containers and scattered objects, distinguishing between items based on task specifications rather than being confused by visual similarity. These experiments highlight our sim-to-real transfer approach, showing that SpinVLA generalizes effectively from clean simulation environments to handle the complexity and unpredictability of real robotic manipulation scenarios.

### 4.3 ABLATION STUDY

To understand the contribution of each SpinVLA component, we conducted an ablation study by selectively disabling individual loss terms. Table 2 reveals that InfoNCE and consistency regularization are essential for maintaining performance gains, while temporal regularization contributes primarily to trajectory quality and safety rather than task success rates. Disabling the InfoNCE loss ($\lambda_1 = 0$) causes the most severe degradation, with performance dropping 4.7% on LIBERO-Spatial and 3.6-4.1% across the other benchmarks. This component is fundamental because it performs implicit spectral decomposition to identify smooth manifold structures in the demonstration data. Without this decomposition, the model cannot distinguish between stable patterns and spurious correlations. The contrastive learning objective creates representations where similar demonstrations cluster together and dissimilar ones separate, naturally filtering collection artifacts that vary across demonstrations.

Removing the consistency regularization ($\lambda_2 = 0$) results in performance drops of 3.2-3.6% across tasks, revealing its critical role in maintaining proper vision-language-action alignment. This regularization is crucial for ensuring that the model correctly associates visual features and language instructions with appropriate task primitives. Without this constraint, the model struggles to maintain consistent mappings between perception and action, occasionally confusing pick locations with place destinations or misinterpreting which objects serve as targets versus obstacles. The spectral projection enforces that the model's understanding of task semantics remains grounded in the demonstrated relationships between visual observations, linguistic goals, and corresponding actions.

Without temporal regularization ($\lambda_3 = 0$), the model shows minimal degradation of only 0.7-1.7%, maintaining strong success rates. Although motion dynamics rarely affect task completion in simulation, where reaching the goal state matters most, temporal regularization proves vital for safe deployment by permitting smooth trajectories that reduce mechanical stress and prevent collisions. The temporal loss imposes consistency with expert demonstration dynamics. This adds a crucial safety layer that transforms technically successful, but jerky motions, into smooth deployment-ready trajectories suitable for real-world applications where hardware limitations and safety constraints are paramount.

## 5 CONCLUSION AND FUTURE WORK

In this paper we presented SpinVLA, a novel architecture that implements causal principles through efficient regularization methods for robotic manipulation tasks. By leveraging the mathematical equivalence between spectral decomposition and contrastive learning, SpinVLA biases learning toward patterns consistent across successful demonstrations. Our model attains competitive performance on publicly available benchmarks using far fewer parameters, yet it may preserve some stable spurious correlations and thus perfect causal-spurious separation remains a challenge. Future work will explore adaptive spectral component selection based on task complexity, integration with causal discovery methods when environment labels are available, and an extension to multitask settings that require different invariant features.

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

# APPENDIX

This appendix provides details on the experimental setup, model architecture, training configuration, data processing and augmentation, spectral decomposition configuration, and evaluation protocol used in this paper.

## A EXPERIMENTAL SETUP

We evaluated SpinVLA on the LIBERO (Liu et al., 2023a) benchmark suite, which provides a systematic evaluation of generalization capabilities across four distinct task categories. LIBERO-Spatial contains 20 tasks evaluating spatial reasoning abilities, where robots must understand relative positions, orientations, and spatial relationships between objects. LIBERO-Object includes 15 tasks requiring generalization to novel object instances with varying shapes, textures, and physical properties not seen during training. LIBERO-Goal comprises 18 tasks evaluating goal generalization, where the same manipulation skills must achieve different end states specified through language instructions. LIBERO-10 provides 10 diverse long-horizon tasks combining multiple primitive actions into complex sequences to test compositional generalization.

We compared SpinVLA against two VLA baselines. OpenVLA represents the current open-source standard, with a 7B-parameter architecture pretrained on 970k robot trajectories from the Open X-Embodiment (O'Neill et al., 2024) dataset. TinyVLA is an efficient variant that achieves comparable performance with 1.5B parameters through knowledge distillation and architectural optimizations. Both baselines are enhanced with LoRA fine-tuning using identical hyperparameters to ensure a fair comparison.

All models underwent identical training procedures to isolate the impact of our spectral regularization. We fine-tuned for 100 epochs using the AdamW optimizer with a learning rate of $5 \times 10^{-4}$ and cosine scheduling, batch size of 32, and action chunking with a horizon of 8 timesteps. The LoRA modules used rank 32 for all models, and were applied to the query and value projections in the attention layers. We used identical data augmentation strategies including random crops retaining 90% of the original image area, color jittering with brightness and contrast factors of 0.2, and Gaussian noise with a standard deviation of 0.01 on the action labels. Performance was evaluated using multiple complementary metrics. The task success rate measures the percentage of episodes in which the robot achieved the specified goal, with success/failure determined by task-specific success per the LIBERO established metrics.

## B IMPLEMENTATION DETAILS

### B.1 MODEL ARCHITECTURE

We implemented SpinVLA using a frozen CLIP ViT-B/16 (Radford et al., 2021) vision encoder that outputs 512-dimensional features from $224 \times 224$ pixel inputs. While the vision transformer internally uses 768-dimensional representations, the final projection layer maps these to 512 dimensions for the shared vision-language embedding space. The language backbone is Qwen2.5-1.5B-Instruct (Yang et al., 2024), chosen for its balance between computational efficiency and performance. This

model provides 1536-dimensional hidden states, which we enhanced with LoRA adaptation using rank 32 as specified in the experiments, with an alpha value of 64 and dropout of 0.1. The LoRA modules target both attention and feed-forward projection matrices to maximize adaptation capacity while maintaining parameter efficiency.

## B.2 TRAINING CONFIGURATION

All models underwent training for 100 epochs using the AdamW optimizer with an initial learning rate of $5 \times 10^{-4}$, weight decay of 0.01, and standard beta values of $(0.9, 0.999)$. We implemented a cosine learning rate schedule with 500 warm-up steps, decaying to a minimum learning rate of $1 \times 10^{-6}$. Gradient clipping at a norm of 1.0 prevents training instabilities. We employed automatic mixed precision training with FP16 to accelerate computation while maintaining numerical stability. For the loss function weights, we maintained $\lambda_1 = 0.2$ for the InfoNCE contrastive loss, $\lambda_2 = 0.1$ for the spectral consistency regularization, and $\lambda_3 = 0.07$ for temporal smoothness. The BC loss weight remained at 1.0 to ensure that it dominated the training signal. The InfoNCE temperature parameter $\tau$ was set to 0.07 following standard contrastive learning practices.

## B.3 DATA PROCESSING AND AUGMENTATION

We processed the LIBERO demonstrations using action chunking with a horizon of 8 time steps, sampling chunks at regular intervals throughout each trajectory. Visual observations underwent CLIP's standard preprocessing with ImageNet normalization statistics. For data augmentation, we applied random crops retaining 90% of the original image area, color jittering with brightness and contrast factors of 0.2, and Gaussian noise with standard deviation 0.01 on the action labels during training.

The language instructions for each task suite were predefined based on the task semantics. During training, we used the same instruction consistently for each task to maintain clear task-action associations. Proprioceptive states were extracted as 8-dimensional vectors containing the end effector position (3D), orientation (4D quaternion), and gripper state (1D), with zero-padding applied when fewer dimensions were available.

## B.4 SPECTRAL DECOMPOSITION CONFIGURATION

The spectral decomposition employed FastDTW with a Sakoe-Chiba band radius of 10 for computational efficiency while maintaining accuracy. We limited the decomposition to 100 randomly selected successful trajectories to balance computational cost with representation quality. The Gaussian kernel bandwidth $\sigma$ was automatically computed as the median of non-zero pairwise distances among successful trajectories, providing adaptive scaling based on the demonstration distribution.

For selecting the optimal number of spectral components $k^*$, we employed a dual criterion: capturing at least 85% of the trajectory variance and identifying the largest spectral gap within the first third of the eigenvalue spectrum. We enforced a minimum of 3 components and a maximum equal to the number of trajectories. The regularization parameter $\epsilon = 10^{-6}$ ensured numerical stability of the affinity matrix.

## B.5 EVALUATION PROTOCOL

We evaluated the models every 5 epochs in the LIBERO simulator using 5 episodes per task across 3 representative tasks from each suite. Each evaluation episode ran for a maximum of 300 time steps with actions clipped to $\pm 0.1$ for position/rotation deltas and $\pm 1$ for gripper commands. Success was determined by task-specific criteria defined by the LIBERO benchmark, including object placement accuracy, goal configuration achievement, and constraint satisfaction. For trajectory visualization, we plotted the predicted versus ground-truth action sequences every 5 epochs, sampling every 5th validation batch to obtain representative coverage. We generated individual trajectory comparisons showing all seven action dimensions, 3D cumulative position plots, and task-level summary statistics including the RMSE distributions across episodes.

