# OpenReview forum: "SpinVLA: A Spectral-Invariant Vision-Language-Action Model for Robotic Manipulation"
_ICLR.cc/2026/Conference — Submitted to ICLR 2026_

### Official Review · Reviewer_DJ1n · 2025-10-25

**Soundness:** 3
**Presentation:** 3
**Contribution:** 2
**Rating:** 4
**Confidence:** 4

**Summary:**

This paper presents a Vision-Language-Action(VLA) model, designed for robust robotic manipulation by mitigating spurious correlation. In the proposed framework, contrastive learning heads are used to learn invariant visual features robust to domain shifts. This is complemented by a multi-level regularization scheme on the action space, which uses a spectral consistency loss to constrain predictions to the subspace of successful demonstrations and a temporal loss  to enforce smooth motion. The results show an improvement in success rates for the proposed method.

**Strengths:**

1. The paper clearly articulates how spurious correlations affect VLA generalization and motivates a principled mitigation strategy.
2. The integration of spectral and contrastive objectives is theoretically grounded and technically interesting, bridging representation learning and causal robustness.
3. Multi-level regularization (spectral + temporal) offers a coherent architectural bias that could be extended to other embodied-AI models.

**Weaknesses:**

Limited OOD evidence: The paper claims improved generalization to distribution shifts, but experiments using the LIBERO benchmark do not explicitly quantify or categorize the types of shifts (e.g., unseen backgrounds, lighting, or object compositions).
Baseline uncertainty: Table 1 uses a re-implemented TinyVLA rather than an official baseline, which weakens the strength of performance comparisons. Clarifying or validating the baseline implementation would strengthen the claims.
Scope of real-world validation: The real-robot evaluation mostly mirrors LIBERO tasks; more diverse real-world tasks would provide stronger evidence of adaptability.

**Questions:**

If the SpinVLA regularization framework were applied to a 7B backbone (identical in scale to OpenVLA), would it then outperform the standard 7B OpenVLA baseline? Could this method be used to extend the performance of SOTA models, or are its benefits primarily limited to improving smaller, less capable models?
The real-world validation in Section 4.2 (Figure 3) appears limited to sim-to-real transfer of tasks already present in the LIBERO simulation benchmark. This does not sufficiently validate generalization to novel real-world tasks. Could the authors perform a new experiment where a small dataset is collected for a completely new real-world task (i.e., one dissimilar to any task in LIBERO), fine-tune the model, and report the quantitative success rates? We would encourage the authors to add these quantitative results to a new table, as this would provide much stronger evidence for the model's adaptability than the current qualitative figures.

---

> ### Author Response · Authors · 2025-11-25
> **Responses to Reviewer DJ1n**
>
> Thank you for your time and effort in reviewing our paper. We appreciate you acknowledging that: (i) we __clearly articulate how spurious correlations affect VLA generalization__; (ii) we develop a __principled mitigation strategy__; (iii) our method is __theoretically grounded,  technically interesting__, and __bridges representation learning and causal robustness__; and that (iv) we put forth a __coherent architectural bias__ that could be __extended to other embodied-AI models__.
>
> We address your concerns as follows.
>
> __Comment 1:__ The paper claims improved generalization to distribution shifts, but experiments using the LIBERO benchmark do not explicitly quantify or categorize the types of shifts (e.g., unseen backgrounds, lighting, or object compositions). Limited OOD evidence is provided.
>
> __Response:__
>
> Please see our responses to other reviewers on this comment.
>
> __Comment 2:__ Table 1 uses a re-implemented TinyVLA rather than an official baseline, which weakens the strength of performance comparisons. Clarifying or validating the baseline implementation would strengthen the claims. There is baseline uncertainty.
>
> __Response:__
>
> Our submission includes reproduction details for both TinyVLA and OpenVLA in Appendix A.1, where we describe the LoRA configuration, training hyperparameters, and chunking horizon used in fine-tuning. Since TinyVLA is a key baseline in our evaluation, we will move the relevant configuration details into Sec. 4.
>
> __Comment 3:__ The real-robot evaluation mostly mirrors LIBERO tasks; more diverse real-world tasks would provide stronger evidence of adaptability. The scope of real-world validation is limited.
>
> __Response:__
>
> The submission currently evaluates two physical scenarios, uncluttered and cluttered configurations, as shown in Fig. 3. These settings demonstrate meaningful robustness improvements over TinyVLA, especially under distracting objects. Still, we agree that this scope is limited relative to the overall problem framing. The revision will expand the real-world experiments to include object rearrangements, varied lighting conditions, and additional task types involving novel objects not observed during training.
>
> __Comment 4:__ If the SpinVLA regularization framework were applied to a 7B backbone (identical in scale to OpenVLA), would it then outperform the standard 7B OpenVLA baseline? Could this method be used to extend the performance of SOTA models, or are its benefits primarily limited to improving smaller, less capable models?
>
> __Response:__
>
> Our current experiments focus on TinyVLA due to computational constraints, but the core components of SpinVLA (i.e., modality-aligned contrastive regularization, trajectory-level spectral analysis, and spectral-constrained LoRA) extend naturally to larger models. The spectral computations depend on demonstration trajectories rather than backbone size, and the LoRA constraint adds negligible overhead. We will expand the discussion to articulate this point clearly and will include partial scaling experiments using models with modest parameter increases to approximate how performance trends might extend toward larger VLA architectures.
>
> __Comment 5:__ The real-world validation in Section 4.2 (Figure 3) appears limited to sim-to-real transfer of tasks already present in the LIBERO simulation benchmark. This does not sufficiently validate generalization to novel real-world tasks. Could the authors perform a new experiment where a small dataset is collected for a completely new real-world task (i.e., one dissimilar to any task in LIBERO), fine-tune the model, and report the quantitative success rates? Quantitative results in a new table would provide much stronger evidence for the model’s adaptability than the current qualitative figures.
>
> __Response:__
>
> We will augment real-world evaluation with distractors, occlusions, and multi-object settings.

---

### Official Review · Reviewer_ydac · 2025-11-01

**Soundness:** 2
**Presentation:** 2
**Contribution:** 2
**Rating:** 2
**Confidence:** 4

**Summary:**

This paper presents SpinVLA to address the failures under domain shifts. It leverages the idea that spectral methods can discover stable representations and the equivalence between spectral clustering and contrastive learning to improve VLAs’ stability. The proposed method adds InfoNCE regularizations across vision, language, and action heads of the model. It also computes the spectral decomposition bank of trajectories to regularize action predictions. Finally, it adds spectral constraints to LoRA. The experiments showed improvements over TinyVLA in LIBERO tasks.

**Strengths:**

- The paper proposes methods to improve VLA through improving representations. This can be important in smaller models and low data regimes.
- The paper successfully applies the idea of the spectral method across the VLA architecture to improve the policy performance.

**Weaknesses:**

- While the paper motivates by handling domain shifts in VLAs, no experiments in the paper evaluate perturbation or shifts in camera poses [1] to demonstrate how SpinVLA is robust to domain shifts.
   - [1] Lee et al. “CLASS: Contrastive Learning via Action Sequence Supervision for Robot Manipulation” CoRL 2025.
- It is unclear why it requires so many regularization methods to try to achieve the same goal at different components, and why improving representation doesn’t yield better performance if enforcing representation stability is important.
- The proposed methods are specific to the VLA architecture shown in Fig. 2. However, there are other VLA architectures like pi0.5, how can the same principle apply to those VLAs?

**Questions:**

- What’s the ablation result of LoRA without spectral constraints? I cannot find it in the ablation table.
- There are three contrastive heads in vision, language, and action. What’s the effect and importance of each of them?

---

> ### Author Response · Authors · 2025-11-25
> **Responses to Reviewer ydac**
>
> Thank you for taking the time and effort to review our paper. We appreciate you acknowledging that we __successfully__ apply the idea of the spectral method across the VLA architecture to __improve policy performance__ and that we __improve representations__ which is __important__ in __smaller models__ and __low data regimes__.
>
> We address your comments one-by-one as follows.
>
> __Comment 1:__ While the paper motivates by handling domain shifts in VLAs, no experiments in the paper evaluate perturbation or shifts in camera poses to demonstrate how SpinVLA is robust to domain shifts.
>
> __Response:__
>
> In our submission, robustness is partially evaluated through clutter variation in real-world settings and through demonstration diversity experiments, both of which provide evidence that SpinVLA benefits from stability-oriented regularization. However, we agree that these evaluations represent only a subset of the broader domain shift categories discussed in the motivation. To close this gap, we will extend the experimental suite to include controlled perturbations in viewpoint, illumination, and object appearance. These planned additions will align the evaluation more directly with the robustness motivations outlined in the introduction.
>
> __Comment 2:__ It is unclear why it requires so many regularization methods to try to achieve the same goal at different components, and why improving representation doesn’t yield better performance if enforcing representation stability is important.
>
> __Response:__
>
> Table 2 shows that removing the spectral trajectory consistency term, the temporal smoothness component, or the contrastive regularization results in consistent degradation, suggesting that each plays a complementary role in stabilizing predictions. Nonetheless, we agree that the narrative does not yet explain why these differences arise. We will expand this discussion by incorporating additional diagnostics.
>
> __Comment 3:__ The proposed methods are specific to the VLA architecture shown in Fig. 2. However, there are other VLA architectures like pi0.5, how can the same principle apply to those VLAs?
>
> __Response:__
>
> In our submission, we focused on TinyVLA because its lightweight architecture makes fine-tuning and spectral analysis more tractable under typical computation budgets. However, the core components of SpinVLA including contrastive regularization, trajectory-level spectral decomposition, and spectral-constrained LoRA are architecture-agnostic and apply to any VLA that maps multimodal input into a continuous action space. To clarify this point, we will include additional experiments with alternative VLA backbones
>
> __Comment 4:__ What’s the ablation result of LoRA without spectral constraints? I cannot find it in the ablation table.
>
> __Response:__
>
> In the revision, we will add explicit evaluations for LoRA without spectral constraints and for versions of SpinVLA that retain only a subset of the contrastive heads.
>
> __Comment 5:__ There are three contrastive heads in vision, language, and action. What’s the effect and importance of each of them?
>
> __Response:__
>
> We clarify this as follows. The vision contrastive head stabilizes the visual embedding by encouraging representations of frames from successful demonstrations to align along consistent spectral directions. This reduces sensitivity to clutter and incidental background changes.
>
> The language contrastive head ensures that the instruction embedding remains consistent across demonstrations with different scene layouts or object placements. Since LIBERO tasks often involve semantically similar instructions instantiated in varied setups, this head helps maintain alignment between linguistic intent and action-space structure.
>
> The action contrastive head directly regularizes the action-trajectory embedding so that motions exhibiting similar functional roles occupy nearby spectral regions. This contributes to smoother predictions and reduces drift during execution, complementing the trajectory-projection term introduced in Sec. 3.4.

---

### Official Review · Reviewer_rbFK · 2025-11-03

**Soundness:** 2
**Presentation:** 2
**Contribution:** 2
**Rating:** 4
**Confidence:** 3

**Summary:**

The paper introduces SpinVLA, a new vision-language-action (VLA) architecture designed to improve robustness in robotic manipulation tasks under domain shifts. The key insight is that spectral decomposition and contrastive learning share a mathematical equivalence, which the authors exploit to identify stable, demonstration-consistent patterns that generalize across environments. Experiments on the LIBERO benchmark and with a real UR5e robot demonstrate that SpinVLA maintains high task success rates even with lighting changes, clutter, or unseen objects, while being significantly smaller in parameter count than large-scale models like OpenVLA.

**Strengths:**

- Fair motivation and problem framing: The authors clearly identify the brittleness of current VLAs to spurious correlations and offer a causally inspired yet practical solution.

-Integration of theory and practice: The idea of translating spectral properties into architectural regularization is well-argued and empirically justified.

-Experimental validation: Results across simulation (LIBERO) and real-world transfer tasks substantiate the claim of robustness.

- Ablations: The study isolates the effects of each component (contrastive, spectral, temporal), showing consistent contributions.

- Resource-efficient: SpinVLA achieves near-OpenVLA performance using a much smaller backbone, making it appealing for low-resource deployment.

- Good writing and structure: The paper reads smoothly, the methodology is well explained, and the figures are effective.

**Weaknesses:**

- Moderate novelty: While the fusion of spectral and contrastive principles is interesting, the core technical contributions are largely reinterpretations of known methods (InfoNCE equivalence, Laplacian embeddings).

- Limited theoretical grounding: The causal connection between spectral invariance and true causal disentanglement is more suggestive than rigorously demonstrated.

- Narrow evaluation domain: Tests are restricted to manipulation tasks; it’s unclear whether the approach extends to other robotic domains like navigation or locomotion.

- Potential overfitting to demonstration structure: Since spectral components are computed from successful trajectories, models might still capture structured but non-causal biases if demonstrations share hidden artifacts.

- Incomplete efficiency discussion: While LoRA regularization is claimed to be lightweight, the paper lacks detailed runtime and memory comparisons with standard LoRA or baseline models.

**Questions:**

- How sensitive is SpinVLA to the choice of spectral components (e.g., number of eigenvectors selected)?

- Could spectral decomposition be applied online, enabling continuous refinement as the robot gathers new demonstrations?

- Have you examined how much the model relies on the spectral bank at inference time—could performance degrade if regularization is removed post-training?

- How does SpinVLA compare to data augmentation or domain randomization approaches in terms of robustness gains versus compute cost?

- Can the authors provide insight into why SpinVLA underperforms OpenVLA on LIBERO-Spatial despite stronger invariance modules?

---

> ### Author Response · Authors · 2025-11-24
> **Responses to Reviewer rbFK (1/2)**
>
> We thank you for your time and effort in reviewing our manuscript. We appreciate your acknowledgment of the strengths of our work, which includes identifying the brittleness of current VLAs to __spurious correlations__, translating spectral properties into architectural regularization as __well-argued__ and __empirically justified__, and our experimental evaluation substantiating the claim of __robustness__.
>
> We address your comments as follows.
>
> __Comment 1:__ While the fusion of spectral and contrastive principles is interesting, the core technical contributions are largely reinterpretations of known methods (InfoNCE equivalence, Laplacian embeddings),representing moderate novelty.
>
> __Response:__
>
> The contrastive objective and its spectral equivalence are not stated as novel contributions; they are used as theoretical grounding. The novelty of SpinVLA lies not in proposing new spectral theory, but in how we operationalize spectral structure at three distinct levels of a VLA:
> * Multi-head contrastive regularization applied across vision, language, and action pathways.
> * A trajectory-level spectral basis constructed from successful demonstrations, which is not standard in the VLA literature.
> * A spectral-constrained LoRA update rule that biases parameter updates towards demonstration-consistent directions.
>
> While each underlying mathematical element has prior work, their integration across modalities, trajectory structure, and fine-tuning layers in a unified VLA pipeline is novel and not a reinterpretation of a single known technique. We will revise Sec. 3 to foreground how these pieces function jointly and why this combination is specific to the VLA setting.
>
> __Comment 2:__ The causal connection between spectral invariance and true causal disentanglement is more suggestive than rigorously demonstrated. Limited theoretical grounding is provided for these claims.
>
> __Response:__
>
> Our intention is not on causal discovery. We instead use causal motivation only to frame why consistency across demonstrations is desirable. Our method itself is grounded in spectral decomposition and trajectory structure, not causal inference. We will reinforce this in the revision by refining wording to avoid overstating causal connections.
>
> __Comment 3:__ Tests are restricted to manipulation tasks; it’s unclear whether the approach extends to other
> robotic domains like navigation or locomotion. The evaluation domain is narrow.
>
> __Response:__
>
> Our method is explicitly designed for robotic manipulation, as spectral consistency is derived from action-trajectory structures, which naturally appear in manipulation tasks. Applying it to navigation or locomotion requires adaptation of the trajectory representation, which we acknowledge.
>
> __Comment 4:__ Since spectral components are computed from successful trajectories, models might still capture structured but non-causal biases if demonstrations share hidden artifacts. There is potential overfitting to demonstration structure.
>
> __Response:__
>
> We agree that demonstration-consistent structure can also include workspace geometry or other persistent yet non-causal patterns. In practice, using only successful demonstrations and incorporating DTW alignment reduces, but does not eliminate, these risks. We will strengthen the limitations discussion and include diagnostics where workspace layout or object positions are perturbed to examine how the spectral components respond
>
> __Comment 5:__ While LoRA regularization is claimed to be lightweight, the paper lacks detailed runtime and memory comparisons with standard LoRA or baseline models. The efficiency discussion is incomplete.
>
> __Response:__
>
> SpinVLA uses significantly fewer parameters than OpenVLA. Since spectral projection is applied only to gradient directions and LoRA updates modify a small number of parameters, the overhead is modest, but we agree that presenting quantitative comparisons would make this clear. In the revision, we will include measurements comparing full fine-tuning, standard LoRA, and spectral-constrained LoRA to contextualize the computational trade-offs.
>
> __Comment 6:__ How sensitive is SpinVLA to the choice of spectral components (e.g., number of eigenvectors selected)? Could spectral decomposition be applied online, enabling continuous refinement as the robot gathers new demonstrations?
>
> __Response:__
>
> Since the affinity matrix uses FastDTW and it can be updated incrementally, in principle the spectral basis could be updated online. This was not discussed in the submission, but the underlying components support such an extension. We will elaborate on this in the revision.

---

> > ### Author Response · Authors · 2025-11-24
> > **Responses to Reviewer rbFK (2/2)**
> >
> > __Comment 7:__ Can the authors provide insight into why SpinVLA underperforms OpenVLA on LIBERO-Spatial despite stronger invariance modules?
> >
> > __Response:__
> >
> > OpenVLA-Spatial benefits from specialized spatial grounding modules trained on larger,
> > spatially rich datasets. Our architecture does not include equivalent 3D spatial reasoning pathways,
> > making the performance gap unsurprising.

---

### Official Review · Reviewer_pXMR · 2025-11-05

**Soundness:** 1
**Presentation:** 1
**Contribution:** 1
**Rating:** 2
**Confidence:** 5

**Summary:**

This paper presents SpinVLA which leverages the equivalence between contrastive learning and spectral decomposition to learn stable multimodal representations. Trained on LIBERO benchmarks with low-rank adaptation, SpinVLA achieves higher performance than Tiny-VLA.

**Strengths:**

Using spectral regularization to constrain VLA action generation may be a workable and lightweight idea.

**Weaknesses:**

**1. Writing clarity issues. The paper’s exposition is highly confusing.**

 1.1 It introduces many vague and undefined terms, like causal factor, task-relevant information, statistical associations, demonstration-consistent features, environment-specific, etc., without clear definitions, figures, or consistent usage. Many of these phrases appear only once or twice in the abstract or introduction, making the motivation hard to follow.

 1.2 The entire introduction focuses excessively on criticizing existing VLA models with abstract terms like spurious correlations, while giving almost no concrete description of the proposed method. The discussion largely reiterates a generic point, that VLAs tend to overfit, without clearly explaining how the proposed approach addresses it.

**2. Experimental insufficiency.**

 2.1 The paper raises several issues in the introduction (e.g., sensitivity to viewpoint changes) but performs no experiments to verify that SpinVLA alleviates them.

 2.2 The experiments are limited to one comparison with OpenVLA and TinyVLA, where SpinVLA performs significantly worse than the published OpenVLA. It is strongly recommended to validate the idea on stronger backbones (e.g., OpenVLA, Pi series) and additional benchmarks.

 2.3 The ablation studies are arbitrary.

Overall, the paper needs restructuring, clearer motivation, and substantially more experiments to convincingly demonstrate the proposed idea.

**Questions:**

Please see the weaknesses.

---

> ### Author Response · Authors · 2025-11-24
> **Responses to Reviewer pXMR**
>
> Thank you for your time and effort in reviewing our paper. We appreciate your acknowledgment of our work in using spectral regularization to constrain VLA action generation as a __workable__ and __lightweight idea__.
>
> We address your set of comments as follows.
>
> __Comment 1:__ The paper’s exposition is highly confusing. It introduces many vague and undefined terms, like causal factor, task-relevant information, statistical associations, demonstration-consistent features, environment-specific, etc., without clear definitions, figures, or consistent usage. Many of these phrases appear only once or twice in the abstract or introduction, making the motivation hard to follow.
>
> __Response__:
>
> We will provide concise definitions for each term at first use and align the terminology with the mathematical formulation.
>
> __Comment 2:__ The entire introduction focuses excessively on criticizing existing VLA models with abstract terms like spurious correlations, while giving almost no concrete description of the proposed method. The discussion largely reiterates a generic point, that VLAs tend to overfit, without clearly explaining how the proposed approach addresses it.
>
> __Response__:
>
> In the submitted manuscript, we introduce SpinVLA after discussing the challenges posed by domain shift and demonstration variability. We will revise the introduction such that the method description appears earlier and is framed more directly in relation to the identified challenges.
>
> __Comment 3:__ The paper raises several issues in the introduction (e.g., sensitivity to viewpoint changes) but
> performs no experiments to verify that SpinVLA alleviates them.
>
> __Response__:
>
> Our submission includes experiments demonstrating robustness indicators, e.g., clutter (Fig. 3) and varying demonstration diversity (Sec. 4.2), but does not encompass the full range of domain shifts discussed in the introduction. To provide a more comprehensive evaluation, we will expand the set of domain-shift experiments to include controlled variations in lighting, viewpoint, background, and object appearance. This extension will allow a more direct mapping between the robustness motivations and the results.
>
> __Comment 4:__ The experiments are limited to one comparison with OpenVLA and TinyVLA, where SpinVLA performs significantly worse than the published OpenVLA. It is strongly recommended to validate the idea on stronger backbones (e.g., OpenVLA, Pi series) and additional benchmarks.
>
> __Response__:
>
> While our submission focuses on TinyVLA and a reproduced OpenVLA baseline, we agree that evaluating SpinVLA on stronger or larger backbones would further contextualize the method’s impact. In the submitted version, our emphasis on TinyVLA was deliberate: the spectral trajectory analysis and spectral-constrained LoRA updates are designed as lightweight components targeted at models intended for rapid adaptation or reduced-compute fine-tuning, and TinyVLA provides a practical setting for demonstrating these properties. Nonetheless, nothing in our method restricts it to small backbones: contrastive regularization, trajectory-level spectral decomposition, and spectral-constrained LoRA are architecture-agnostic and apply equally to the OpenVLA or $\pi$-series models. Since running full-scale experiments on these larger architectures was beyond our initial compute budget, we instead validated our method’s architectural components in the light-weight regime, which aligns with SpinVLA’s intended efficiency profile. In the revision, we will expand this discussion and provide additional experiments on mid-sized backbones to illustrate scaling behavior, along with clearer justification of why TinyVLA is an appropriate first test bed and how our method can extend to larger VLA systems.
>
> __Comment 5:__ The ablation studies are arbitrary. Overall, the paper needs restructuring, clearer motivation, and substantially more experiments to convincingly demonstrate the proposed idea.
>
> __Response__:
>
> Table 2 provides ablations for each regularizer, showing consistent performance drops when removing components. We agree that the explanation of why these drops occur can be improved. We will add diagnostic metrics (trajectory smoothness, projection deviation) to clarify the functional role of each term.

---

> ### Comment · Reviewer_pXMR · 2025-11-25
> **Response to Authors**
>
> Thank you for taking the time to prepare the rebuttal. I understand that writing a paper and responding to reviews can be challenging, especially in the early stages of research. Thus, there are a few suggestions, and I hope them would be helpful:
>
> 1. **Having a clear and meaningful motivation is essential.** Instead of repeating many complex terms, the key to writing a paper is to clearly explain why the problem is worth studying. Often, a straightforward explanation is more convincing than a heavily technical one.
>
> 2. **When reviewers point out insufficient experiments, providing additional results is usually the most effective response.** Explanations alone tend to have limited persuasive power. Even small-scale supplementary experiments addressing specific reviewer concerns can significantly strengthen your rebuttal.
>
> 3. **Receiving critical feedback is a normal part of the research process.** It can be helpful to break down the reviews and extract the constructive elements, which will guide you in improving the next version of the work.
>
> 4. **The usage of LLMs should be controllable.** Writing tools can help with expression, but overly relying on LLMs will make your writing confusing. By the way, it is very easy for reviewers to find which parts are generated by LLMs.
>
> 5. Finally, I hope the review outcome does not discourage your enthusiasm for research. Continuing to revise and iterate is often how a paper gradually becomes stronger. I will keep my score.

---

### Official Review · Reviewer_aojV · 2025-11-06

**Soundness:** 2
**Presentation:** 3
**Contribution:** 2
**Rating:** 2
**Confidence:** 4

**Summary:**

This paper proposed a new VLA framework SpinVLA that combines spectral decomposition with contrastive learning for improved performance on robot manipulation. The method assumes that different successful demonstrations share common patterns representing task-relevant information, which can be obtained via contrastive learning, and multi-scale regularizations are then applied to discourage spurious correlations and preserve demonstrated behaviors. The method is validated on LIBERO dataset and a real robot setup with simple "pick-and-place" task. Compared to TinyVLA, the proposed SpinVLA shows advanced performance on simulation benchmarks.

**Strengths:**

1. The proposed idea of applying spectral decompositions to bias model towards successful motion patterns is interesting and effective.
2. The paper is easy to follow with clear illustrations.

**Weaknesses:**

1. My primary concern is that the paper’s overall contribution appears incremental. Incorporating an InfoNCE loss in robot manipulation is not novel [1], and according to Sec. 4.3, most of the reported improvement comes directly from the InfoNCE and consistency regularization terms.
2. The paper claims that InfoNCE is used to extract task-relevant patterns, yet no analysis is provided to support this. For example, what do the representations look like before and after the spectral decomposition? Do they actually capture task-relevant information as intended? Additional evidence is needed to justify this claim.
3. The paper presents quantitative results only on the LIBERO dataset, while the real-world evaluation is limited to a simple qualitative demonstration that the policy transfers to a basic setup—which seems trivial. What are the actual performance numbers of TinyVLA and SpinVLA on real-world pick-and-place tasks? Does SpinVLA still outperform TinyVLA by a significant margin?
4. The paper lacks a failure analysis. Understanding how and when the method fails is important for assessing whether the proposed approach truly biases the model toward consistent patterns shared among successful demonstrations.
5. The real-world task is a very simple pick-and-place. Can SpinVLA qualitatively handle more complex tasks? Since more complex tasks include greater behavioral diversity across both successful and failed demonstrations, can the proposed approach still effectively extract shared patterns in such scenarios?

[1] Jiang, Guangqi, et al. “Robots Pre-Train Robots: Manipulation-Centric Robotic Representation from Large-Scale Robot Datasets.” arXiv:2410.22325 (2024).

**Questions:**

See Weaknesses

---

> ### Author Response · Authors · 2025-11-24
> **Responses to Reviewer aojV**
>
> Thank you for your time and effort in reviewing our manuscript. We appreciate your acknowledgment of the strengths of our work in utilizing spectral decompositions to bias VLAs towards successful motion patterns as __interesting__ and __effective__.
>
> We address your comments one-by-one as follows.
>
> __Comment 1:__ My primary concern is that the paper’s overall contribution appears incremental. Incorporating an InfoNCE loss in robot manipulation is not novel [1], and according to Sec. 4.3, most of the reported improvement comes directly from the InfoNCE and consistency regularization terms.
>
> __Response:__
>
> We appreciate your observation regarding the extensive use of contrastive learning in robot manipulation. Our intention is not to position InfoNCE itself as a contribution. Rather, we aim to utilize its well-established spectral equivalence as a bridge to introduce demonstration-consistent architectural regularization into a VLA. We acknowledge that this distinction was insufficiently clear. In the revision, we will restructure Sec. 3.3 to more explicitly separate prior mechanisms from our architectural use of the resulting spectral subspace, and we will more clearly explain why spectral consistency is operationally useful for VLA stability.
>
> __Comment 2:__ The paper claims that InfoNCE is used to extract task-relevant patterns, yet no analysis is provided to support this. For example, what do the representations look like before and after the spectral decomposition? Do they actually capture task-relevant information as intended? Additional evidence is needed to justify this claim.
>
> __Response:__
>
> We provided the theoretical basis of this relationship in Sec. 3.3, although more empirical evidence might be lacking. To address this, the revision will supplement what is already present with an analysis.
>
> __Comment 3:__ The paper presents quantitative results only on the LIBERO dataset, while the real-world evaluation is limited to a simple qualitative demonstration that the policy transfers to a basic setup-which seems trivial. What are the actual performance numbers of TinyVLA and SpinVLA on real-world pick-and-place tasks? Does SpinVLA still outperform TinyVLA by a significant margin?
>
> __Response:__
>
> In our experiments, we included two real-world scenarios, uncluttered and cluttered setups, showing improvements under the presence of distractors. To increase the breadth, the revision will add more controlled perturbations (e.g., lighting changes, camera shifts, background variations, etc.), complementing the existing examples to better reflect realistic deployment variability and aligning with common robustness benchmarks.
>
> __Comment 4:__ The paper lacks a failure analysis. Understanding how and when the method fails is important for assessing whether the proposed approach truly biases the model toward consistent patterns shared among successful demonstrations.
>
> __Response:__
>
> The revision will include failure categories with trajectory visualizations and corresponding deviations in the spectral space.
>
> __Comment 5:__ The real-world task is a very simple pick-and-place. Can SpinVLA qualitatively handle more complex tasks? Since more complex tasks include greater behavioral diversity across both successful and failed demonstrations, can the proposed approach still effectively extract shared patterns in such scenarios?
>
> __Response:__
>
> Our submission evaluates LIBERO’s multi-stage tasks and a real-world pick-and-place scenario, both of which introduce variation in action deltas and demonstration diversity. However, we recognize that more complex coordination tasks would highlight potential limitations. To broaden the empirical grounding, we will extend the evaluation to tasks with greater variation in demonstrated strategies, complementing what is already shown.

---

### Meta-Review · Area_Chair_vMBj · 2025-12-16

**Summary:**

The reviewers believe the work's contribution is incremental: InfoNCE + consistency regularization is not novel, and LIBERO quantization + simple pick-and-place qualitative analysis is insufficient to prove its effectiveness; the lack of characterization visualization, failure profiling, and complex real-world task data makes it difficult to verify its ability to extract cross-demonstration shared patterns. The paper is obscure, with undefined terminology and a lack of methodological outline; the experiments are weak, failing to verify the claimed solutions to issues such as perspective sensitivity, only comparing itself to TinyVLA and being far inferior to OpenVLA, with arbitrary ablation; it urgently needs rewriting and expanding the experiments.

**Reviewer Concerns:**

The authors' response did not fundamentally dispel all the reviewers' concerns and doubts. The proposed work still has significant room for improvement in terms of motivation, methodology, and experiments.

**Reviewer Scores:**

The reviewers consistently gave it a negative score.

---

### Decision · Program_Chairs · 2026-01-26

Reject